# Combined Impact of Omicron Vaccination and Environmental Risk Exposure: A Thailand Case Study

**DOI:** 10.3390/vaccines11020297

**Published:** 2023-01-29

**Authors:** Weerawat Ounsaneha, Orapin Laosee, Thunwadee Tachapattaworakul Suksaroj, Cheerawit Rattanapan

**Affiliations:** 1Faculty of Science and Technology, Valaya Alongkorn Rajabhat University under the Royal Patronage, Pathumthani 13180, Thailand; 2ASEAN Institute for Health Development, Mahidol University, Nakhonpathom 73710, Thailand

**Keywords:** COVID-19, booster dose, environmental risk exposure, Omicron variants

## Abstract

This research aimed to determine the levels of COVID-19 booster dose vaccinations in Thai populations in areas with environmental risk exposure during the Omicron outbreak. Five of twenty provinces in Thailand were selected by assessing environmental risk exposure for study settings. A total of 1038 people were interviewed by a structured questionnaire. The predicting factors of COVID-19 booster dose vaccinations were analyzed by univariate and multivariate analysis. The results showed that 69.4% (95% CI 66.5–72.1) of the population was vaccinated with COVID-19 booster doses. Multiple logistics regression revealed that the female gender (AOR 1.49, 95% CI 1.11–2.00), all age groups from 38 to 60 years old, all education levels of at least secondary school, high income (AOR 1.16, 95% CI 1.15–2.24), populations having experience with COVID-19 infection (AOR 2.27, 95% CI 2.05–3.76), knowledge of vaccine (AOR 1.78, 95% CI 1.11–2.83), and trusting attitude (AOR 1.76, 95% CI 1.32–2.36) were factors among those more likely to take COVID-19 booster dose vaccinations in high-environmental-risk-exposure areas. Therefore, an effective booster dose campaign with education programs to increase attitudes toward booster vaccinations should be implemented for the resilience of COVID-19 prevention and control.

## 1. Introduction

Various countries around the world have been concerned about the largest public health crisis since the first confirmed case of a novel coronavirus disease 2019 (COVID-19) in December 2019 [1]. Over 300 million people with COVID-19 infections and 5.5 million deaths have been reported due to the ongoing global pandemic [2]. Effective approaches including quarantine, social distancing, and home isolation have been implemented to reduce viral transmission. However, public health and safety issues must be questioned with these approaches [3]. Hence, the COVID-19 vaccine approach with development, implementation, and uptake has been proposed for the prevention of the pandemic. Worldwide vaccine uptake is a successful and effective mechanism for COVID-19 outbreak control [4].

Shekhar et al. [5] found that vaccination against Severe Acute Respiratory Syndrome Coronavirus 2 (SARS-CoV-2) was the principal preventive measure against the spread of the virus and the development of severe COVID-19 diseases. An important tool for infection control, hospitalization reduction, and death from the COVID-19 pandemic has been focused on vaccines. However, global concern about the side effects of COVID-19 vaccines has arisen because of insufficient knowledge of the short- and long-term effects, distrust of vaccine companies, and the belief that the virus is not harmless [6]. Messenger RNA (mRNA) technology has been developed as the novel vaccine platform for decreasing the severity and transmissibility of SARS-CoV-2 [7]. Specifically, high efficacy values of 4.1% and 95% and 100% and 89% for COVID-19 protection against severe symptoms were found in the mRNA vaccine brands of the Pfizer/BioNtech BNT162b2 and Moderna mRNA-1273 vaccines, respectively. Other vaccination platforms with viral-vector-based vaccines, such as Johnson & Johnson and Astra Zeneca, have been reported to have high efficacy against COVID-19 [8]. From these findings, COVID-19 vaccine development, implementation, and uptake have urgently been recommended as an effective approach to COVID-19 control [8].

The onerous effects, including economic crisis, psychological attack, and lost social interaction, were ongoing during the global COVID-19 pandemic [9,10]. The mutation variants of COVID-19 were formally announced as the following: Alpha (α), Beta (β), Gamma (γ), Delta (δ), and Omicron. In December 2021, a total of 352 confirmed cases in 27 countries were reported [11]. The increased transmission and immune evasion were found in the Omicron variant, and the re-infection rate in this variety was three times more than α, β, γ, and δ [12]. Many reports [13,14] have found that more than 38% of the effectiveness of BNT162b2 against SARS-CoV-2 infection was reduced with the SARS-CoV-2 Omicron variant. Whether a booster dose of COVID-19 vaccine is necessary has become a point of debate [5]. Feikin et al. [15] reported that the protective reduction in COVID-19 vaccine effectiveness was found in the overtime period. In the end of 2022, over half the population of the world had taken at least two doses of mRNA vaccine against the virus, and the third dose of the vaccine is currently underway worldwide. At the end of 2021, over 70% effective infection control was found in the conventional COVID-19 variants such as B.1.617.2 (Delta) or B.1.1.7 (Alpha) [16,17]. The second dose of the vaccine against COVID-19 has maintained vaccine effectiveness for at least 6 months [18].

The Universal Coverage Scheme (UCS) has been implemented in Thailand since 2002, enhancing the total health expenditure from 63 to 77% and reducing out-of-pocket expenses from 27 to 12% [19]. Approximately 76% of Thais (approximately 47 million people) were covered by this scheme [20]. The National Health Security Board in 2021 [21] reported 47.74 million (99.61%) Thais were registered at healthcare sectors under the UCS. The total disbursement obligation was USD 4419.52 million (101.33% of the UHC budget, excluding healthcare providers’ salaries, totaling USD 4361.67 million), which was used to commission healthcare services. Of all the impacts of the three waves of COVID-19 between March 2020 and August 2021, healthcare service systems in Thailand were the most affected [22]. The Thai government promoted the policy of 100 million doses of COVID-19 vaccine for 2021 and 120 million doses for 2022 to achieve 70% coverage for its citizens and accounted for booster vaccines effective for the new variants [14].

For vaccine uptake in Thailand, 75%, 65%, and 18% of the population have taken a first dose, a second dose booster, and a third dose booster of COVID-19 vaccines, respectively. The Thai Ministry of Public Health (MOPH) reported and confirmed the first case of the Omicron variant in early December 2021. This situation caused significant concern for the government regarding the capacity of the Thai healthcare system to handle the new variant outbreak [23]. In Thailand, two groups of vaccines are fully registered and approved by the Ministry of Public Health for emergency use: (1) free-of-charge vaccines including Sinovac, Pfizer, and AstraZeneca and (2) pay-by-yourself vaccines including Sinopharm and Moderna [24]. Side effects of COVID-19 vaccination uptake such as pain, fatigue, swelling at the injection site, fever, headache, muscle and joint aches, dizziness, nausea, low blood pressure, shortness of breath, rapid heartbeat, nasal congestion, and thromboembolism were reported and monitored [25]. Hence, the Thai government has promoted the procurement of vaccine uptake with safe access and effective vaccines [26]. Particularly, such data evidence the need for additional COVID vaccine booster doses in the period of Omicron predominance.

Bontempi et al. [27] stated that the diffusion patterns in the pandemic situation came from environmental, economic, and social dimensions. The distribution of COVID-19 infection rapidly increased death cases in cities with social interaction and geo-environmental factors such as low wind and frequently high air pollution levels [28,29]. Specifically, Axiotakis et al. [30] implied that public locations with close social contact and public areas were significantly influenced by the Omicron variant. The population density, frequent activities, and the people’s transportation in urban areas were important issues for finding sustainable epidemic and environmental risk control. Some significant environmental factors, including humidity, atmospheric temperature, and ventilation filter systems in hotels, hospitals, or houses in close contact with the population influenced the Omicron variant [30]. In addition, Sohail et al. [9] found that the virus distribution was limited by certain conditions, specifically 6 g/kg of humidity and an average air temperature above 51 F. Coccia [31] identified that the environmental exposure risk assessment of the city or pandemic areas for preventing COVID-19 exposure was limited. The reduction in COVID-19 infection rates in high-risk-level areas with environmental exposure was highlighted and challenged.

During data collection in this study, there was an Omicron variant wave of the COVID-19 outbreak in Thailand, and over 25 thousand COVID-19 cases were admitted to the hospital, with more than eighty deaths/day reported during the highest epidemic peak in Thailand from 27 March to 2 April 2022 [32]. Vaccine efficacy on the reduction in virus spreading and safe resumption of public and social activities were the main reasons for recommendation of booster vaccination uptake. Moreover, the high immunogenicity against the Omicron variant of the booster dose reduced the risk of infection [33]. Data from the Ministry of Public Health, Thailand on 2 April 2022 reported that only 34.4% of Thai people had received the third dose of vaccination [32]. Specifically, the potential risk of infectious diseases with the actual risk given by infected individuals and deaths of COVID-19 was found to be at a high level in environmental risk exposure areas, as implied by the significant spreading factors of the Omicron variant [30,31].

Therefore, the objective of this study was to determine the predictive factors of COVID-19 for booster dose vaccinations among the population in environmental risk exposure areas of Thailand during the Omicron variant outbreak. The environmental risks of exposure in areas of Thailand were investigated and adapted using the measurement by Mario Coccia [31]. These findings can propose the appropriate promotion strategy for booster dose vaccination that supports policymakers to understand the behavior pattern of COVID-19 protection in the areas of environmental risk exposure for reducing the impact of public health and economic issues caused by the pandemic.

## 2. Materials and Methods

### 2.1. Study Setting

The study setting in this research was selected from five provinces with the highest areas of environmental risk exposure level in Thailand. Coccia [31] identified that the factors used in the assessment of the environmental risk of exposure consisted of air pollution, wind speed, population density, and respiratory disorders of people. Hence, the sources of information and factors used for assessing the environmental risk of exposure areas in this study are presented in Table 1. The procedure for assessment of environmental risk of exposure level is as follows. Step (1) selects the 20 Thai provinces with the highest cases of COVID-19 infection in Thailand on 7 March 2022, based on data by the Ministry of Public Health (MOPH) [32]. Step (2) cllects the data of environmental factors (Table 1) including air pollution, wind speed, population density, and respiratory disorders for the 20 provinces from Step (1). Step (3) calculates the 25th, 50th, and 75th percentiles for data values from Step (2) of four environmental factors in the 20 provinces and categorizes the percentile values in four sets: Set 1 (lower than the 25th percentile), Set 2 (between the 25th and 50th percentiles), Set 3 (between the 50th and 75th percentiles), and Set 4 (greater than the 75th percentile). Step (4) assigns the scores from Step (3) in the 20 provinces with a point value between 0 and 3, where Set 1 = 0, Set 2 = 1, Set 3 = 2, and Set 4 = 3. Then, the data collected from Steps (1), (2), (3), and (4) were calculated as the environmental risk of exposure level (EREL) to COVID-19 infection with using Equation (1) below:(1)$$\mathrm{EREL}=\frac{\left[F1\left(p_{k}\right){+F2\;(p}_{k}\right)+F3\;\left(p_{k}\right){+F4\;(p}_{k}{)]}_{i}}{12}$$
where EREL = environmental risk of exposure level, F1, F2, F3, F4 = factors used to calculate the environmental risk of exposure level to COVID-19 in a given province (F1 = air pollution, F2 = wind speed, F3 = population density, and F4 = respiratory disorders), and P_k_ = score of a province assigned to each factor F_i_, with values between 0 and 3.

The value of the environmental risk of exposure level ranged from 1 to 0. A value close to 1 implies a high environmental risk of exposure area to COVID-19, and a rank close to 0 indicates a low environmental risk of exposure area to COVID-19.

### 2.2. Population and Data Collection Procedure

The survey by face-to-face interviews was conducted on the populations in five provinces with the highest levels of environmental risk exposure from the past. The sample size of participants was calculated using a confidence interval of 95% and an acceptable error of 20. Permission for data collection was approved by the health officers in each area. According to this method, 1038 people were recruited as subjects in this study. The participants were included with the following eligibility criteria: aged over 18 years, Thai citizens, and living in the study setting for more than 6 months. The proportional random sampling by gender and age group was conducted to be representative of citizens of Thailand. Participants voluntarily agreed to enroll in this survey and take part in the research; they provided written informed consent after receipt of the recruitment procedure and research participation instructions provided by a trained research assistant. Anonymity and confidentiality of data were strictly maintained. During data collection, a research assistant followed the government policy for COVID-19 prevention and control. This research was approved by the Committee for Research Ethics (Social Science), Faculty of Social Sciences and Humanities, Mahidol University with certificate approval number 2022/033.2802 and MU-SSIRB number 2022/35(B2).

### 2.3. Questionnaire

The questionnaire as the measurement tool in this study was a modified version of a previous survey [38,39,40]. The pre-test process of the questionnaire was undertaken by 30 populations in another province for testing the validity and reliability of the measurement tool, with a 0.75 Cronbach’s alpha coefficient. After the piloting process, the three parts relating to the objective goal of this study were the following: (1) socio-demographic status (age, gender, level of education, marital status, job, income, and COVID-19 infection history); (2) COVID-19 knowledge of infection in general, preventive behavior, and vaccination domains with 1 point for correct answers and 0 points for incorrect answers; (3) COVID-19 attitude towards risk perception and trust domains using Likert scales scored as Strongly Agree = 4, Agree = 3, Not Sure = 2, Disagree = 1, and Strongly Disagree = 0; (4) COVID-19-preventive behavior along with questions by Likert scales scored as Always = 4, Mostly = 3, Sometimes = 2, Rarely = 1, and Never = 0. A number of those receiving booster dose vaccinations were interviewed with two questions and correctly rechecked by the online vaccine certificate in the Thai formal application on a mobile phone.

### 2.4. Statistical Analysis

Descriptive statistics including percentages, means or medians, standard deviations, or quartile deviations were used for data analysis in this study. A univariate analysis, the Chi-square test, was used to identify the variables associated with COVID-19 booster dose vaccinations. A strong predictor for COVID-19 booster dose vaccination with *p* < 0.05 was identified for multivariate logistics regression. An odds ratio of association was performed by the confidence intervals of logistic regression. SPSS software version 21 was used to calculate all the statistical analyses, and the analysis was conducted at a 5% significance level.

## 3. Results

### 3.1. Area of Environmental Risk of Exposure in Thailand

Table 2 shows the assessment level of environmental risk exposure in the 20 areas with the highest cases of COVID-19 infection in Thailand on 7 March 2022. The study areas, including twenty provinces with the highest cases of COVID-19 infections in Thailand, are shown in Figure 1. The levels of environmental risk exposure in provinces are presented as ranks between 0 and 1, and the rankings of the provinces are also from the highest risk of exposure (1.00) and lowest risk of exposure (0.17). Coccia [31] mentioned that a value of environmental risk exposure higher than 0.75 identified a high risk of exposure to COVID-19. Based on the results, five provinces in the central part of Thailand including Bangkok, Samut Prakan, Samut Sakhon, Nonthaburi, and Pathum Thani were selected as the sample areas for determining COVID-19 booster dose vaccinations among the population during the Omicron variant outbreak.

### 3.2. Population Characteristics

Table 3 presents the population characteristics in this study; 1038 people participated in the areas with the highest environmental risk of exposure during the Omicron variant outbreak. The percentages of the population in each province were similar in terms of ratio. Among the populations, 59.7% were female and 40.3% were male. Most (26.2%) were in the 25 to 37 age group, and a few (8.6%) were 18 to 24 years old. More than 29% of the population had completed secondary school. The majority of the population (55.5%) were married. For occupation, 35.9% of the population were general employees. The majority of the population (57.0%) had a monthly income of more than USD 367. About 43.2% of the population had previously had COVID-19, and family members were the sources of transmission.

### 3.3. COVID-19 Booster Dose Vaccination

A summary of COVID-19 booster dose vaccinations among populations in the Thai areas with the highest environmental risk exposure is presented in Table 4. The results from the survey found that most populations that participated in this survey had received the COVID-19 vaccination. The number of vaccinations were first dose (2.5%), second dose (28.1%), third dose (52.9%), and fourth dose (16.5%). From this finding, the majority of the population (69.4%) reported that they were vaccinated against COVID-19 with a booster dose.

### 3.4. Knowledge, Attitudes, and Preventive Behaviors toward COVID-19

Table 5 shows the level of COVID-19 knowledge, attitude, and preventive behavior in populations in Thai areas with the highest environmental risk exposure. According to the COVID-19 knowledge, three domains of knowledge including general, preventive measurement, and vaccination were measured by a true–false test. The results showed that 62.7% and 90.8% of the population presented a high level of knowledge in general and vaccination domains, respectively. In contrast, 77.4% of the population was found to have low knowledge of the preventive measurement domain. In terms of COVID-19 attitudes, the domains of risk perception and trust were categorized by Likert scale items. Most populations had a good level of COVID-19 attitude, with 57.1% risk perception and 61.8% trust. Additionally, COVID-19 preventive behaviors of the population with Likert scale items resulted in remarkably good (77.6%) and low levels (22.4%) for preventive behaviors.

### 3.5. Logistic Regression Model for COVID-19 Booster Dose Vaccination

The prediction factors for COVID-19 booster dose vaccinations in the areas with the highest environmental risk exposure during the Omicron variant outbreak were identified using multiple and logistic regression analyses within the interest variables from the bivariate analysis with a p-value of 0.05. The results are shown in Table 6. For the bivariate analysis, a significant association was found between COVID-19 booster dose vaccination and the age groups of 25–37 years (OR 1.95, 95% CI 1.19 to 3.20), 38 to 45 years (OR 2.15, 95% CI 1.29 to 3.59), 46–53 years (OR 1.77, 95% CI 1.08 to 2.89), and 54+ years (OR 2.07, 95% CI 1.23 to 3.51). Populations with a diploma degree (OR 1.90, 95% CI 1.23 to 2.94) and a bachelor’s degree or higher (OR 1.58, 95% CI 1.06 to 2.34) for education levels were significantly more likely to take the booster dose vaccination. For the population’s monthly income, more than USD 154 was found to be a significant indicator for booster dose vaccination (OR = 2.22, 95% CI = 1.70 to 2.91). Positive COVID-19 compared with negative COVID-19 was significantly associated with booster dose vaccination (OR = 2.22, 95% CI = 1.68 to 2.87). Populations with a high knowledge of vaccination (OR = 2.10, 95% CI = 1.37 to 3.22) were significantly more likely to have booster dose vaccinations. Additionally, populations with good COVID-19 attitudes on risk perception and trust domains were significantly more likely to take booster dose vaccinations than those with poor COVID-19 attitudes.

The predictive factors associated with COVID-19 booster dose vaccination were investigated with a multivariate logistic regression analysis. The results showed that female sex (AOR 1.49, 95% CI 1.11 to 2.00) predicted the COVID-19 booster dose vaccination uptake among the population at the highest level of the environmental risk exposure areas. Being in the age groups of 38 to 45 years (AOR 2.00, 95% CI 1.12 to 3.59), 46 to 53 years (AOR 2.07, 95% CI 1.15 to 3.74), and 54+ years (AOR 2.83, 95% CI 1.50 to 5.34) contributed to COVID-19 booster dose vaccination. The education levels of secondary school (AOR 1.16, 95% CI 1.05 to 2.47), diploma degree (AOR 2.11, 95% CI 1.28 to 3.47), and bachelor’s degree or higher (AOR 2.84, 95% CI 1.74 to 4.62) were associated with COVID-19 booster dose vaccinations. Populations with monthly income levels >USD 154 (AOR 1.16, 95% CI 1.15 to 2.24) were more willing to receive the COVID-19 booster doses. In addition, the COVID-19 booster doses were significantly higher in the populations testing positive for COVID-19 (AOR 2.27, 95% CI 2.05 to 3.76). Only COVID-19 knowledge on vaccination (AOR 1.78, 95% CI 1.11 to 2.83) and COVID-19 attitudes on trust (AOR 1.76, 95% CI 1.32 to 2.36) were significantly associated with the uptake of COVID-19 booster doses.

## 4. Discussion

The Omicron variant of the COVID-19 epidemic outbreak also caused rapid transmission with populations in many countries including the United States [41], the United Kingdom [42], and South Africa [43]. In the context of Thailand, Suphanchaimat et al. [23] investigated the peak and daily death number of COVID-19 Omicron variant incidents. The low rate of COVID-19 vaccination was 49,523 cases by day 73 and over 270 by day 50, respectively. This study also confirmed the coverage of vaccinations to attack the Omicron variant. One-third of the peak incident cases in the worst-case scenario were reduced by a speedy vaccination rate. Hence, a program of booster dose vaccinations in Thailand should be implemented to prevent severe disease and death in the Thai people during an Omicron outbreak. Specifically, the area of high environmental risk exposure presented a significant influence via the Omicron variant [30]. In the current study, the COVID-19 booster dose vaccination of Thai people in the area of environmental risk exposure during the Omicron outbreak was determined.

The levels of environmental risk exposure in highly endemic provinces of Thailand were assessed by environmental factors adapted from Coccia [31], including air pollution, wind speed, population density, and respiratory disorders. The results identified that five provinces in Bangkok metropolitan areas had the highest levels of environmental risk exposure to a viral agent because of the high concentration of air pollution in the pattern of PM 10, which can be supported by the longer period and diffusion ability of COVID-19 outdoors as well as expanded human transmission in public areas [27,28]. The province with the highest level of environmental risk exposure in this study was Samut Sakhon, which was caused by the high total days exceeding the limits set for PM 10, low wind speed, high density of population, and high rates of mortality for lung cancer (data not shown). Consistent with the report from Yamsrual et al., [44] found that Samut Sakhon populations indicated that the main environmental problems in their perception were air and water pollution and odors. From these findings, the prevention and control of environmental pollution in Thai provinces can reduce the potential risk of COVID-19 transmission. The proposed concept of environmental risk exposure can promote an up-to-date policy that supports policymakers to understand the knowledge gap in infectious disease transmission and protect from a new wave of COVID-19 as well as new viral agents in the potential risk areas of Thailand.

The current survey found that nearly 70% of populations in the areas with environmental risk exposure took COVID-19 booster dose vaccinations. This uptake rate of COVID-19 vaccination was higher compared with the findings from a current systematic review and meta-analysis in the world setting on COVID-19 booster dose vaccination by Abdelmoneim et al. [45]. The uptake rate of the COVID-19 booster dose vaccination from eight publications around the world with 12,995 participants was 31%. In addition, only 28.8% of the population in Bangkok received the booster dose from September to December 2021. During the data collection period (March 2022–June 2022), it was found that the uptake rate of COVID-19 booster doses in the Bangkok Metropolitan Region in this study was 2 times higher than the previous report 3 months later [46] because the risk of transmission and safe activity resumption in the public area were elicited by taking COVID-19 booster vaccinations during the rise of the Omicron variant [33]. The policy of COVID-19 booster doses among the entire population was strongly supported by the Thai government [25]. Hence, the areas with high environmental risk exposure and high incidences of COVID-19 infection were the priority targets for vaccination coverage in Thailand.

The results from multiple logistics regression analysis found that the groups of population characteristic factors including female gender, all age groups from 38–60 years, all education levels of at least secondary school, and high income were predictors of COVID-19 booster dose vaccination uptake among the populations with the highest levels of environmental risk exposure in Thailand. Some research [47,48] identified that there were no significant differences found between males and females in terms of COVID-19 vaccine acceptance. In the present study, females had more COVID-19 vaccinations than males. This finding was consistent with the report among the global population [49], global dental students [50], and Palestinian dental students [51]. Urrunaga-Pastor Diego et al. [52] reported that the anti-vaccine groups in the female gender had lower exposure than the male. In addition, Thai females were more likely to engage in preventive health behaviors against COVID-19 infection than males [53]. Hence, the uptake of COVID-19 booster dose vaccination was employed by Thai women during the time of this research survey. In terms of age group, being aged between 25 and 54 years old was a strong predictive factor for COVID-19 vaccination booster dose because these people were more willing to accept and follow the recommendations of the vaccine policy [41]. Consistent with our findings in this study, the age groups of 38–45 years, 46–53 years, and over 54 years in the studied populations were found to predict COVID-19 booster dose vaccination. However, the results of knowledge in the present study found that populations with a higher level of education (at least secondary school) exhibited more COVID-19 booster dose vaccinations because broad information about the side effects of vaccinations would be screened and analyzed by educated population groups [54]. Regarding the role of economic status as a determinant of COVID-19 booster vaccination, the high-income population groups were more likely to accept the booster doses because having lower-income status may cause hesitation to take booster doses because of losing income due to illness from the side effects of vaccines [55]. Not surprisingly, those having experience with COVID-19 infections were significantly less likely to take COVID-19 booster dose vaccinations. Various studies [56,57] implied that population groups having experience with COVID-19 infections were more likely to accept the COVID-19 vaccination. Al-Hatamleh Mohammad A. I. et al. [58] reported that many symptoms, including general fatigue, headache, fever, anosmia, and cough, were the most frequent health burdens of COVID-19 infection. Populations were willing to take the COVID-19 booster dose vaccinations for reducing the suffering of COVID-19 infections [59].

The current study found that a strong predictor of COVID-19 booster vaccination among populations in the areas with environmental risk exposure was knowledge level concerning vaccination. Consistent with the previous research in Thailand [53] and Italy [60], it was identified that the high knowledge of respondents on vaccines was significantly associated with vaccination acceptance and uptake against COVID-19, because individual experience towards the learning process would change health and self-care behaviors [61]. Comprehensive knowledge of COVID-19 infections may induce people to accept vaccinations to protect themselves and their family members [62]. Finally, COVID-19 attitude on trust predicted an enhancement of COVID-19 booster dose vaccinations [63], implying that trust in COVID-19 vaccinations is associated with the acceptance rate of COVID-19 booster dose vaccinations. For the populations with good attitudes, the level of awareness could be increased by having an appropriate attitude during a pandemic [64].

This study highlighted the need to develop a specific recommendation for the enhancement of booster vaccinations in high-risk areas for infection transmission in the context of environmental exposure. The significant benefits on environmental, public health, social, and economic issues in cities can be enhanced by pollution reduction, and the risk of transmission in cities can be reduced for any new wave or future pandemics of COVID-19 [28]. In addition, this study contributes to the understanding of how to increase motivation in the unvaccinated groups in high-risk areas. Knowledge of booster doses and trusting attitudes were the main factors for increasing the uptake rate of COVID-19 booster doses in these areas. Hence, an effective booster dose campaign with an education program on effectiveness and safety to increase attitude levels on booster vaccination should be implemented to increase the COVID-19 booster dose vaccinations throughout the country and increase the resilience performance of COVID-19 prevention and control [65].

## 5. Conclusions

The recommendation for booster dose vaccination uptake was a global approach to COVID-19 outbreak prevention during the Omicron variant. The rapid spread of COVID-19 infections in a city was stimulated by social interaction as well as the environmental risk exposure factor. Thus, the widespread distribution of COVID-19 booster dose vaccination was provided by the Thai government during that crisis. The findings from this survey determined the crucial knowledge on prediction factors for the booster dose of the COVID-19 vaccination in the high-risk areas of Thailand in the environmental exposure context. To sum up, good knowledge of booster dose vaccination and a good attitude with trust were found to be strong predictors of COVID-19 booster dose vaccination among populations in the areas with a high risk of environmental exposure. Therefore, an educational campaign on the effectiveness and safety of vaccinations was still demanded to expand public attitudes toward COVID-19 booster dose vaccinations [46] and was urgently needed in Thailand [47]. From this result, the findings imply that the potential risk of transmission with the pollution reduction in cities can be deduced for the incident rate of future COVID-19 pandemics.

## Figures and Tables

**Figure 1 vaccines-11-00297-f001:**
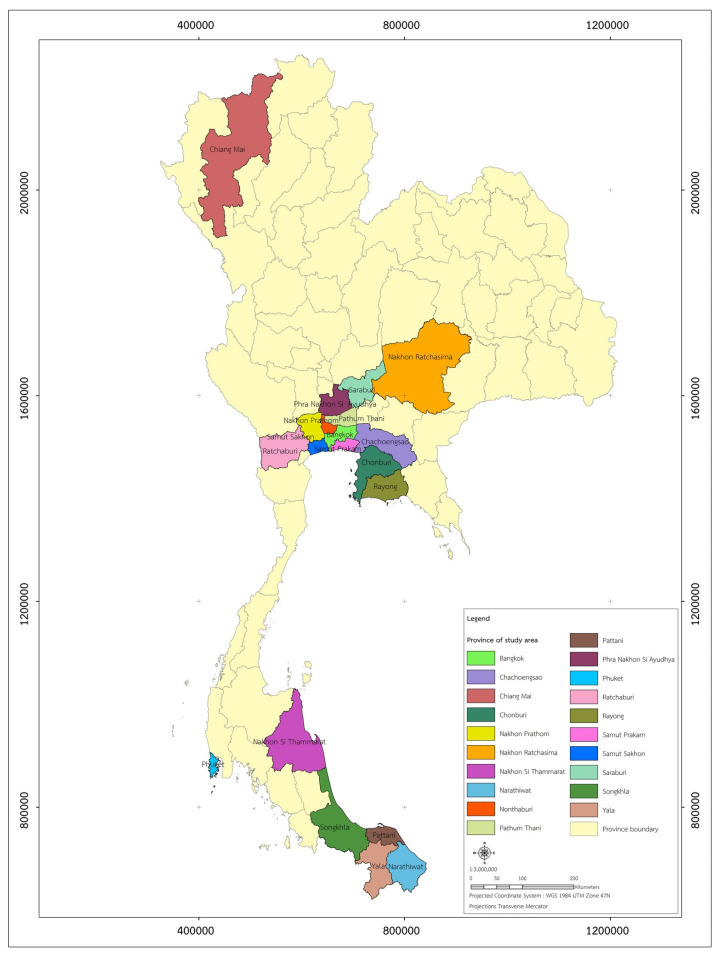
Location of twenty provinces with the highest cases of COVID-19 infections in Thailand, 2022.

**Table 1 vaccines-11-00297-t001:** Factors used for COVID-19 diffusion determination in an urban area.

Items	Factor Item	Data Collection	Sources
1	Air pollution	Total days exceeding the limits set for PM 10 in 2021	Pollution Control Department [34]
2	Wind speed	Wind speed km/h in 2021	Thai Meteorological Department [35]
3	Population density	Density of population, inhabitants per km^2^ in 2021	National Statistical Office of Thailand [36]
4	Respiratory disorders	Rates of mortality for lung cancer per 100,000 people in 2021	Health Information System Development Office [37]

**Table 2 vaccines-11-00297-t002:** Number cases of COVID-19 infection, total value of environmental factors, and level of environment risk exposure in twenty provinces of Thailand.

Order	Provinces	Number of Cases of COVID-19 Infection	Total Value of Environmental Factors	Level of Environmental Risk Exposure
* 1	Bangkok	587,121	10.00	0.83
* 2	Samut Prakan	186,846	9.00	0.75
3	Chon Buri	162,627	6.00	0.50
* 4	Samut Sakhon	131,566	12.00	1.00
* 5	Nonthaburi	120,711	9.00	0.75
6	Songkhla	76,184	5.00	0.42
7	Nakhon Si Thammarat	74,140	3.00	0.25
8	Rayong	62,822	4.00	0.33
* 9	Pathum Thani	62,333	9.00	0.75
10	Ratchaburi	61,111	8.00	0.67
11	Pattani	53,159	1.00	0.08
12	Nakhon Ratchasima	56,743	7.00	0.58
13	Nakhon Pathom	53,196	5.00	0.42
14	Yala	52,918	2.00	0.17
15	Chachoengsao	51,037	6.00	0.50
16	Phuket	48,433	7.00	0.58
17	Chiang Mai	48,218	7.00	0.58
18	Phra Nakhon Si Ayutthaya	46,892	8.00	0.67
19	Narathiwat	45,609	3.00	0.25
20	Saraburi	44,642	5.00	0.42

Note: * = Five provinces of high environmental risk exposure for data collection.

**Table 3 vaccines-11-00297-t003:** Population characteristics (*n* = 1038).

Characteristic	Category	Number, (%)
Area	Bangkok	224 (21.6)
	Samut Prakan	197 (19.1)
	Samut Sakhon	165 (15.9)
	Nonthaburi	243 (23.4)
	Pathum Thani	209 (20.1)
Sex	Male	418 (40.3)
	Female	620 (59.7)
Age	18–24	89 (26.2)
	25–37	226 (21.8)
	38–45	263 (25.3)
	54+	188 (18.1)
Education	Primary school	169 (16.3)
	Secondary school	307 (29.6)
	Diploma degree	193 (18.6)
	Bachelor’s degree or higher	369 (35.5)
Marital status	Single	393 (37.9)
	Married	576 (55.5)
	Divorce	69 (6.6)
Occupation	Self-employed	205 (19.7)
	General employee	373 (35.9)
	Student	44 (4.2)
	Government sector	118 (11.2)
	Private sector	245 (23.6)
	Farmer	17 (1.6)
	None	38 (3.7)
Monthly income (USD)	Less than USD 154	168 (16.2)
	USD 154.1–367	278 (26.8)
	More than USD 367	592 (57.0)
Have you had COVID-19 infection before?	Yes	448 (43.2)
	No	590 (56.8)
Source of COVID-19 infection	Do not know	131 (12.6)
	Family member	145 (14.0)
	Colleague	96 (9.6)
	High-risk area	70 (6.7)
	Other	1 (0.1)

**Table 4 vaccines-11-00297-t004:** Number of COVID-19 booster dose vaccinations.

Item	Category	Number (%)
Number of vaccines received	1	26 (2.5)
	2	292 (28.1)
	3	549 (52.9)
	4	171 (16.5)
Booster dose received	Yes	720 (69.4)
	No	318 (30.6)

**Table 5 vaccines-11-00297-t005:** Knowledge, attitude, and preventive behaviors towards COVID-19.

Item	Category	Percentage (%)
**COVID-19 Knowledge Level**		
**General**		
Low	387	37.3
High	651	62.7
**Preventive measurement**		
Low	803	77.4
High	235	22.6
**Vaccination**		
Low	95	9.2
High	973	90.8
**COVID-19 attitude level**		
**Risk perception**		
Poor	445	42.9
Good	513	57.1
**COVID-19 attitude level**		
**Trust**		
Poor	396	38.2
Good	642	61.8
**COVID-19 Preventive level**		
Poor	232	22.4
Good	806	77.6

**Table 6 vaccines-11-00297-t006:** Logistic regressions model for COVID-19 booster dose vaccination.

Variable	Booster Dose Vaccination		COR ^a^ (95% CI) ^c^	*p*-Value	AOR ^b^ (95% CI) ^c^	*p*-Value
	Yes (%)	No (%)				
**Sex**						
Male	35.4	64.6	1		1	
Female	27.4	72.6	1.45 (1.11−1.89)	0.006	1.49 (1.11−2.00)	0.008
**Age**						
18−24	44.9	55.1	1		1	
25−37	29.4	70.6	1.95 (1.19−3.20)	0.007	1.63 (0.93−2.86)	0.083
38−45	27.4	72.6	2.15 (1.29−3.59)	0.003	2.00 (1.12−3.59)	0.019
46−53	31.6	68.4	1.77 (1.08−2.89)	0.023	2.07 (1.15−3.74)	0.015
54+	28.2	71.8	2.07 (1.23−3.51)	0.006	2.83 (1.50−5.34)	0.001
**Education**						
Primary school	43.8	56.2	1		1	
Secondary school	38.1	61.9	1.26 (0.83−1.85)	0.227	1.16 (1.05−2.47)	0.029
Diploma degree	24.5	75.5	1.90 (1.23−2.94)	0.004	2.11 (1.28−3.47)	0.003
Bachelor’s degree or Higher	25.7	74.3	1.58 (1.06−2.34)	<0.001	2.84 (1.74−4.62)	<0.001
**Marital status**						
Single/divorced	29.0	71.0	1.14 (0.88−1.50)	0.307	1.31 (0.94−1.81)	0.100
Married	31.9	68.1	1		1	
**Monthly income (USD)**						
<367	40.0	59.6	1		1	
>367	23.3	76.7	2.22 (1.70−2.91)	<0.001	1.16 (1.15−2.24)	0.005
**COVID-19 infection**						
No	40.2	59.8	1			
Yes	23.4	76.6	2.2 (1.68−2.87)	<0.001	2.77 (2.05−3.76)	<0.001
**COVID-19 Knowledge**						
**General**						
Low	32.3	67.7	1			
High	29.6	70.4	1.13 (0.86−1.48)	0.630		
**Preventive measurement**						
Low	31.0	69.0	1			
High	29.4	70.6	1.08 (0.78−1.48)	0.121		
**Vaccination**						
Low	46.3	53.7	1		1	
High	28.2	71.8	2.1 (1.37−3.22)	0.001	1.78 (1.11−2.83)	0.015
**COVID-19 attitude**						
**Risk perception**						
Poor	25.9	74.1	1			
Good	25.3	74.7	1.64 (1.25−2.14)	<0.001		
**Trust**						
Poor	39.4	60.6	1		1	
Good	25.2	74.8	1.92 (1.47−2.52)	<0.001	1.76 (1.32−2.36)	<0.001
**COVID-19 Preventive behavior**						
Poor	35.8	64.2	1			
Good	29.2	70.8	1.34 (0.99−1.84)	0.054		

Note: ^a^ COR = crude odds ratio, ^b^ AOR: adjusted odds ratio, and ^c^ 95% CI: confidence interval of 95%.

## Data Availability

Not applicable.

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
