# Peer review of "Combined Impact of Omicron Vaccination and Environmental Risk Exposure: A Thailand Case Study"

_vaccines, 2023, doi:10.3390/vaccines11020297_

Round 1

Reviewer 1 Report

The report is attached.

Author Response

Revision

Reviewer 1

There are no details provided about how the environmental factors are linked with the

omicron.

Author added the link between environmental factors with omicron in line 106.

Which vaccines are developed from 2020 till date for Omicron? Again, the title doesn't match with the contents.

Author changed the title of manuscript for matching with the scope of research in line 1.

I cannot see detailed analysis of data for omicron, for instance check the work:

https://doi.org/10.1016/j.rinp.2022.105300

https://link.springer.com/article/10.1007/s11063-022-10834-5

Author added the finding from https://doi.org/10.1016/j.rinp.2022.105300 and https://link.springer.com/article/10.1007/s11063-022-10834-5

to introduction part in line 55 and 59.

Improve the introduction systematically and take inspiration from the above-mentioned articles. Cites the mentioned works in a paragraph, where you establish the concept of Omicron as the schematics provided there will help to improve your work.

Author added the background of Omicron for improving and taking the inspiration in the manuscript in line 55 -60.

The formula provided in your manuscript must be utilized and the results generated must be graphically presented, the sources must be carefully cited and the data repositories must be acknowledged with consent.

Author carefully rechecked the source of formular and the table of result in the manuscript revision.

Reviewer 2

In the abstract appears information that is not found in the results section of the article.

Author added all results in abstract in the manuscript.

“The results presented that 69.4  % of the population was vaccinated with COVID-19 booster doses.  Multiple logistics regression revealed that the female gender, all age groups from 38 to 60 years old, all education levels of at least secondary school, high-income, populations having experience with COVID-19 infection, 1 knowledge of vaccine and trusting attitude were factors among those more likely to take COVID19 booster dose vaccinations in high environmental risk exposure areas”  If an information appears in the abstract, it should be developed in  with detail in the article.

Author added the all results in abstract the manuscript in line 230-312.

In the abstract the phrase “that 69.4 % of the population “should include the interval of 95%.

Author added the interval of 95% of 69.4 % in line 16.

In the abstract when talking about factors related to taking COVID19 booster dose vaccinations in high environmental risk exposure area, the OR should be included with its 95% confidence interval

Author added the interval of 95% of 69.4 % in line 18-22.

Introduction. : In the introduction briefly describe the health system of Thailand (financing, coverage, benefits, accessibility), with special emphasis on Public Health, and vaccinations this is important to frame the context and facilitate the understanding of the article.

Author added the brief information health system and vaccine policy in Thailand in introduction part in line 71-82

The numbering of the tables should be revised.  There are two Tables 1. One on page 4, and one on the line

Author revised the table based on the reviewer recommendation in line 184.

In Table 1 (page 4) there is an asterisk *, “Wind Speed*” but it is not explained what it means.  It should be explained at the bottom of the page.

Author deleted the * before Wind speed in line 184.

If possible, enter Table 1 (page 4) on the same page where it is first cited for ease of reading.

Author moved table 1 after item 2.1 in line 168.

In Table 1 (page 5) there are several provinces with an asterisk (*1 Bangkok) but it is not explained what it means.  It should be explained at the bottom of the page.

Author added the definition of * in the table bottom in line 227.

The Material and Methods section is unclear.  It should be explained in more detail to allow a reader to reproduce it.

First in paragraph “Step (2) calculates the percentiles of 25thth, 50thth and 75th in each province and categorizes the percentile values in four sets included” What variables are percentiles calculated from?  Do they refer to the values within each province or to the values of the whole country?

Author added the more clarification of step 2 in methodology in line 148-155.

What are the factors F1, F2, F3, and F4?  They should be identified

Author add the factor information of F1, F2, F3, and F4 in line 161.

In summary.   Lines 119 to 139, the entire paragraph should be rewritten providing more information so that any reader understands how it has been prepared.

Author revised the assessment of environmental risk of exposure level for more understand and preparation in line 148-155.

It is indicated that a questionnaire was used, and its psychometric characteristics are provided.  (Chronbach’s alpha).  It would be interesting to include as additional material the English translation of the questionnaire used.

Author added the questionnaire in additional material.

In the material and methods section, it is indicated that Descriptive statistics, including percentages, means or medians, standard deviations 175 or quartile deviations, were used for data analysis in this study were calculated.  This information does not appear in the article.  The phrase should be included or deleted.

Author added the result in material and method in line 230-312.

In the material and methods section, it is indicated that In the material and methods section it is indicated that In the material and methods section it is indicated that univariate analysis as “the Chi-square test was used to identify the variables associated with COVID-19 booster dose vaccinations.  A strong predictor for COVID-19 booster dose vaccination with p<0.05 was identified for multivariate logistics regression.  An odds ratio of association was per- formed by the confidence intervals of logistic regression.”

Author added the result in material and method in line 230-312.

In the results section of the manuscript, there is no chi-square, nor are the results of logistic regression presented.  The information referred to in material and methods should be included in the results, or the paragraph should be deleted from material and methods.

Author added the result in material and method in line 230-312.

The discussion includes data that have not been previously presented in the results, as well as “agents in the potential risk areas of Thailand.

Author added the result in material and method in line 230-312.

The current survey found that nearly 70% of populations in the areas with environ-“ mental risk exposure took COVID-19 booster dose vaccinations.  The data discussed in the discussion should have been presented in the results. meet the objectives of the article.

Author added the result in material and method in line 230-312.

Reviewer 2 Report

The article is interesting because it could provide information on vaccinations against COVID-19 in risk areas.  However, the article does not meet the expectations it generates.  The article is incomplete and does not provide the information that, according to its objectives, should be given; it is as if it was missing pages by accident.  The results section is very brief and does not present the necessary information to meet the objectives of the article.

Abstract

In the abstract appears information that is not found in the results section of the article.

“The results presented that 69.4  % of the population was vaccinated with COVID-19 booster doses.  Multiple logistics regression revealed that the female gender, all age groups from 38 to 60 years old, all education levels of at least  secondary school, high-income, populations having experience with COVID-19 infection, 1 knowledge of vaccine and trusting attitude were factors among those more likely to take COVID19 booster dose vaccinations in high environmental risk exposure areas”  If an information appears in the abstract, it should be developed in  with detail in the article.

In the abstract the phrase “that 69.4 16 % of the population “should include the interval of 95%.

In the abstract when talking about factors related to taking COVID19 booster dose vaccinations in high environmental risk exposure area, the OR should be included with its 95% confidence interval

Introduction. : In the introduction briefly describe the health system of Thailand (financing, coverage, benefits, accessibility), with special emphasis on Public Health, and vaccinations this is important to frame the context and facilitate the understanding of the article.

The numbering of the tables should be revised.  There are two Tables 1. One on page 4, and one on the line

In Table 1 (page 4) there is an asterisk *, “Wind Speed*” but it is not explained what it means.  It should be explained at the bottom of the page.

If possible, enter Table 1 (page 4) on the same page where it is first cited for ease of reading.

In Table 1 (page 5) there are several provinces with an asterisk (*1 Bangkok) but it is not explained what it means.  It should be explained at the bottom of the page.

The Material and Methods section is unclear.  It should be explained in more detail to allow a reader to reproduce it.

First in paragraph “Step (2) calculates the percentiles of 25thth, 50thth and 75th in each province and categorizes the percentile values in four sets included” What variables are percentiles calculated from?  Do they refer to the values within each province or to the values of the whole country?

What are the factors F1, F2, F3, and F4?  They should be identified

In summary.   Lines 119 to 139, the entire paragraph should be rewritten providing more information so that any reader understands how it has been prepared.

It is indicated that a questionnaire was used, and its psychometric characteristics are provided.  (Chronbach’s alpha).  It would be interesting to include as additional material the English translation of the questionnaire used.

In the material and methods section, it is indicated that Descriptive statistics, including percentages, means or medians, standard deviations 175 or quartile deviations, were used for data analysis in this study were calculated.  This information does not appear in the article.  The phrase should be included or deleted.

In the material and methods section it is indicated that In the material and methods section it is indicated that In the material and methods section it is indicated that univariate analysis as “the Chi-square test was used to identify the variables associated with COVID-19 booster dose vaccinations.  A strong predictor for COVID-19 booster dose vaccination with p<0.05 was identified for multivariate logistics regression.  An odds ratio of association was per- formed by the confidence intervals of logistic regression.”

In the results section of the manuscript, there is no chi-square, nor are the results of logistic regression presented.  The information referred to in material and methods should be included in the results, or the paragraph should be deleted from material and methods.

The discussion includes data that have not been previously presented in the results, as well as “agents in the potential risk areas of Thailand.

The current survey found that nearly 70% of populations in the areas with environ-“ mental risk exposure took COVID-19 booster dose vaccinations.  The data discussed in the discussion should have been presented in the results.

Author Response

Reviewer 1

There are no details provided about how the environmental factors are linked with the

omicron.

Author added the link between environmental factors with omicron in line 106.

Which vaccines are developed from 2020 till date for Omicron? Again, the title doesn't match with the contents.

Author changed the title of manuscript for matching with the scope of research in line 1.

I cannot see detailed analysis of data for omicron, for instance check the work:

https://doi.org/10.1016/j.rinp.2022.105300

https://link.springer.com/article/10.1007/s11063-022-10834-5

Author added the finding from https://doi.org/10.1016/j.rinp.2022.105300 and https://link.springer.com/article/10.1007/s11063-022-10834-5

to introduction part in line 55 and 59.

Improve the introduction systematically and take inspiration from the above-mentioned articles. Cites the mentioned works in a paragraph, where you establish the concept of Omicron as the schematics provided there will help to improve your work.

Author added the background of Omicron for improving and taking the inspiration in the manuscript in line 55 -60.

The formula provided in your manuscript must be utilized and the results generated must be graphically presented, the sources must be carefully cited and the data repositories must be acknowledged with consent.

Author carefully rechecked the source of formular and the table of result in the manuscript revision.

Reviewer 2

In the abstract appears information that is not found in the results section of the article.

Author added all results in abstract in the manuscript.

“The results presented that 69.4  % of the population was vaccinated with COVID-19 booster doses.  Multiple logistics regression revealed that the female gender, all age groups from 38 to 60 years old, all education levels of at least secondary school, high-income, populations having experience with COVID-19 infection, 1 knowledge of vaccine and trusting attitude were factors among those more likely to take COVID19 booster dose vaccinations in high environmental risk exposure areas”  If an information appears in the abstract, it should be developed in  with detail in the article.

Author added the all results in abstract the manuscript in line 230-312.

In the abstract the phrase “that 69.4 % of the population “should include the interval of 95%.

Author added the interval of 95% of 69.4 % in line 16.

In the abstract when talking about factors related to taking COVID19 booster dose vaccinations in high environmental risk exposure area, the OR should be included with its 95% confidence interval

Author added the interval of 95% of 69.4 % in line 18-22.

Introduction. : In the introduction briefly describe the health system of Thailand (financing, coverage, benefits, accessibility), with special emphasis on Public Health, and vaccinations this is important to frame the context and facilitate the understanding of the article.

Author added the brief information health system and vaccine policy in Thailand in introduction part in line 71-82

The numbering of the tables should be revised.  There are two Tables 1. One on page 4, and one on the line

Author revised the table based on the reviewer recommendation in line 184.

In Table 1 (page 4) there is an asterisk *, “Wind Speed*” but it is not explained what it means.  It should be explained at the bottom of the page.

Author deleted the * before Wind speed in line 184.

If possible, enter Table 1 (page 4) on the same page where it is first cited for ease of reading.

Author moved table 1 after item 2.1 in line 168.

In Table 1 (page 5) there are several provinces with an asterisk (*1 Bangkok) but it is not explained what it means.  It should be explained at the bottom of the page.

Author added the definition of * in the table bottom in line 227.

The Material and Methods section is unclear.  It should be explained in more detail to allow a reader to reproduce it.

First in paragraph “Step (2) calculates the percentiles of 25thth, 50thth and 75th in each province and categorizes the percentile values in four sets included” What variables are percentiles calculated from?  Do they refer to the values within each province or to the values of the whole country?

Author added the more clarification of step 2 in methodology in line 148-155.

What are the factors F1, F2, F3, and F4?  They should be identified

Author add the factor information of F1, F2, F3, and F4 in line 161.

In summary.   Lines 119 to 139, the entire paragraph should be rewritten providing more information so that any reader understands how it has been prepared.

Author revised the assessment of environmental risk of exposure level for more understand and preparation in line 148-155.

It is indicated that a questionnaire was used, and its psychometric characteristics are provided.  (Chronbach’s alpha).  It would be interesting to include as additional material the English translation of the questionnaire used.

Author added the questionnaire in additional material.

In the material and methods section, it is indicated that Descriptive statistics, including percentages, means or medians, standard deviations 175 or quartile deviations, were used for data analysis in this study were calculated.  This information does not appear in the article.  The phrase should be included or deleted.

Author added the result in material and method in line 230-312.

In the material and methods section, it is indicated that In the material and methods section it is indicated that In the material and methods section it is indicated that univariate analysis as “the Chi-square test was used to identify the variables associated with COVID-19 booster dose vaccinations.  A strong predictor for COVID-19 booster dose vaccination with p<0.05 was identified for multivariate logistics regression.  An odds ratio of association was per- formed by the confidence intervals of logistic regression.”

Author added the result in material and method in line 230-312.

In the results section of the manuscript, there is no chi-square, nor are the results of logistic regression presented.  The information referred to in material and methods should be included in the results, or the paragraph should be deleted from material and methods.

Author added the result in material and method in line 230-312.

The discussion includes data that have not been previously presented in the results, as well as “agents in the potential risk areas of Thailand.

Author added the result in material and method in line 230-312.

The current survey found that nearly 70% of populations in the areas with environ-“ mental risk exposure took COVID-19 booster dose vaccinations.  The data discussed in the discussion should have been presented in the results. meet the objectives of the article.

Author added the result in material and method in line 230-312.

Round 2

Reviewer 1 Report

The report is attached.

Author Response

Reviewer 1

1. Please reduce the length of the title. I would like to suggest the authors a title based on my experience:

Combined Impact of Omicron Vaccination and Environmental Risk Exposure: A Thailand Case Study

Author revised our manuscript title base on reviewer suggestion in line 1.

2. An important acknowledgement missing is the data source, that is missing and must be provided.

Author add the acknowledgement of data source in line 446.

Reviewer 2 Report

The authors have done a great deal of work, which has improved the manuscript, but a final effort is still needed to correct some minor aspects.

 In lines 73 and 75, Thais should be thais.

When TBAT is presented, please also put an equivalence in US Dollars so a reader can easily have an understanding of the magnitude of the amount (lines 76 and 77, line 138)

In line 109, explain  (between brackets) the acronym pf the first time used.

In line 146, “Collect collects” the Word is repeated. Please correct.

Line 190  Change “For multivariate logistic regression analysis, the predictive factors associated with  COVID-19 booster dose vaccination were investigated.”      into “the predictive factors associated with COVID-19 booster dose vaccination were investigated with a multivariate logistic regression analysis”

 In the header of tabla 6, there are two superscripts, a, and b, along with COR and AOR, respectively, but it is not explained what they are. The meaning of superscripts should be put at the foot of the table.

Author Response

Reviewer 2

In lines 73 and 75, Thais should be thais.

Author revised Thais should be thais base on reviewer suggestion in line 72 and 74.

When TBAT is presented, please also put an equivalence in US Dollars so a reader can easily have an understanding of the magnitude of the amount (lines 76 and 77, line 138)

Author revised equivalence in US Dollars base on reviewer suggestion in line 76, 77, 229, 237, 282, 299 and 308

In line 109, explain (between brackets) the acronym pf the first time used.

Author delete pf in the passage in line 108

In line 146, “Collect collects” the Word is repeated. Please correct.

Author delete collect in the passage in line 145

Line 190 Change “For multivariate logistic regression analysis, the predictive factors associated with COVID-19 booster dose vaccination were investigated.”  into “the predictive factors associated with COVID-19 booster dose vaccination were investigated with a multivariate logistic regression analysis”

Author revised the sentence base on the reviewer suggestion in line 290.

In the header of tabla 6, there are two superscripts, a, and b, along with COR and AOR, respectively, but it is not explained what they are. The meaning of superscripts should be put at the foot of the table.

Author add the a, b and c superscripts in the title bottom in line 308 and 315.